# Metabolomic Analysis of Human Astrocytes in Lipotoxic Condition: Potential Biomarker Identification by Machine Learning Modeling

**DOI:** 10.3390/biom12070986

**Published:** 2022-07-15

**Authors:** Daniel Báez Castellanos, Cynthia A. Martín-Jiménez, Andrés Pinzón, George E. Barreto, Guillermo Federico Padilla-González, Andrés Aristizábal, Martha Zuluaga, Janneth González Santos

**Affiliations:** 1Departamento de Nutrición y Bioquímica, Facultad de Ciencias, Pontificia Universidad Javeriana, Bogotá 110311, Colombia; baez.daniel@javeriana.edu.co (D.B.C.); andres_aristizabal@javeriana.edu.co (A.A.); 2Division of Neuropharmacology and Neurologic Diseases, Yerkes National Primate Research Center, Atlanta, GA 30329-4208, USA; cmart51@emory.edu; 3Laboratorio de Bioinformática y Biología de Sistemas, Universidad Nacional de Colombia, Bogotá 111321, Colombia; ampinzonv@unal.edu.co; 4Department of Biological Sciences, University of Limerick, V94 T9PX Limerick, Ireland; george.barreto@ul.ie; 5Royal Botanic Gardens, Kew, London TW9 3AE, UK; federicopg@gmail.com; 6Escuela de Ciencias Básicas Tecnologías e Ingenierías, Universidad Nacional Abierta y a Distancia, Dosquebradas 661001, Colombia; martha.zuluaga@unad.edu.co

**Keywords:** astrocytes, lipotoxicity, neurodegenerative diseases, obesity, metabolomics

## Abstract

The association between neurodegenerative diseases (NDs) and obesity has been well studied in recent years. Obesity is a syndrome of multifactorial etiology characterized by an excessive accumulation and release of fatty acids (FA) in adipose and non-adipose tissue. An excess of FA generates a metabolic condition known as lipotoxicity, which triggers pathological cellular and molecular responses, causing dysregulation of homeostasis and a decrease in cell viability. This condition is a hallmark of NDs, and astrocytes are particularly sensitive to it, given their crucial role in energy production and oxidative stress management in the brain. However, analyzing cellular mechanisms associated with these conditions represents a challenge. In this regard, metabolomics is an approach that allows biochemical analysis from the comprehensive perspective of cell physiology. This technique allows cellular metabolic profiles to be determined in different biological contexts, such as those of NDs and specific metabolic insults, including lipotoxicity. Since data provided by metabolomics can be complex and difficult to interpret, alternative data analysis techniques such as machine learning (ML) have grown exponentially in areas related to omics data. Here, we developed an ML model yielding a 93% area under the receiving operating characteristic (ROC) curve, with sensibility and specificity values of 80% and 93%, respectively. This study aimed to analyze the metabolomic profiles of human astrocytes under lipotoxic conditions to provide powerful insights, such as potential biomarkers for scenarios of lipotoxicity induced by palmitic acid (PA). In this work, we propose that dysregulation in seleno-amino acid metabolism, urea cycle, and glutamate metabolism pathways are major triggers in astrocyte lipotoxic scenarios, while increased metabolites such as alanine, adenosine, and glutamate are suggested as potential biomarkers, which, to our knowledge, have not been identified in human astrocytes and are proposed as candidates for further research and validation.

## 1. Introduction

Obesity is a public health problem affecting a large part of the world’s population. According to the World Health Organization (WHO), in 2016, more than 1.9 billion adults (≥18 years) were overweight, of which around 650 million were obese, resulting in at least 2.8 million deaths worldwide [1]. Obesity is therefore considered a syndrome of multifactorial etiology characterized by an excessive accumulation and release of fatty acids (FA) in adipose and non-adipose tissue to the extent that it is associated with a high risk to health [2]. The excess of FA generates a metabolic condition known as lipotoxicity, which triggers pathological cellular and molecular responses leading to dysregulation of homeostasis and decreases in cell viability [3,4]. Consequently, different molecules that generate inflammatory phenomena can be expressed in the brain and different tissues [5]. Recent studies have shown that a diet rich in simple sugars and saturated fatty acids reduces the expression of brain-derived neurotrophic factor (BDNF), where a decrease in its levels has been associated with metabolic syndrome and cognitive impairment [6,7,8]. In this context, palmitic acid (PA) is the main saturated fatty acid obtained from the diet, constituting approximately 60% of them [9]. Its high consumption in the diet leads to the modification of blood lipid concentrations; for example, it has recently been shown that the acute elevation of blood lipid levels produces an activation of the pro-inflammatory NF-kB pathway, resulting in the increased expression of several pro-inflammatory cytokines, including TNF-α, IL1, and IL6, among others [10,11]. Although PA may result in astrocyte activation [12], neuronal inflammation, and demyelination [13], the direct damage caused by this FA to astrocytes remains relatively unexplored.

To date, and after decades of continuous research, the biological and/or environmental mechanisms underlying the development of the main NDs are still unknown. It has been proposed that the accumulation of different intracellular or extracellular metabolites to toxic levels may trigger the activation of pathological and inflammatory pathways, leading to dysfunction and death phenomena in different brain cells, such as neurons and astrocytes [14,15]. In addition, it has been confirmed that there is a significant metabolic relationship between these two types of cells, where astrocytes functionally regulate neuronal activity, provide trophic and antioxidant support to neurons, facilitate synaptic connectivity and neurotransmission, and counteract eventual metabolic and inflammatory insults [16]. Astrocytes are the most abundant glial cell type in the central nervous system (CNS), representing approximately 50% of the volume of the human brain [17]. For a long time they were considered to be supporting components of nerve cells; yet, as of today it is known that the structure and function of the brain is fundamentally determined by astrocytes [18]. These cells are responsible for the maintenance of neuronal function and the regulation of cerebral blood flow, helping to conserve glycogen for use as an energy source, initiating extracellular ion regulation, participating in synaptic plasticity, producing and degrading neurotransmitters, and assisting in endothelial differentiation. Additionally, astrocytes also play a protective role against the toxicity of reactive oxygen species (ROS), hydrogen peroxide (H_2_O_2_), and metals, since they have specific enzymatic systems to eliminate these compounds [16,17,19,20,21,22,23,24,25,26,27,28]. These structurally complex and highly fibrous glial cells can be understood as the dynamic regulators of neuronal production and its activity [29]. On the other hand, it is important to emphasize that astrocytes also participate in humoral immune responses because they express pattern recognition receptors (PRRs), which are important for the primary recognition of the insult. This interaction activates a series of cytokines that may be involved in mediating immune responses within the CNS [23]. However, and despite its importance, there is still much to understand about its metabolic and functional characteristics, especially when considering pathophysiologies such as AD and PD, among others.

Metabolomics is a multidisciplinary approach facilitating the large-scale detection of genetically- and non-genetically-encoded metabolites of a cellular system under physiological conditions and exposed to different biological contexts, such as drugs or diseases [30,31]. Accordingly, the metabolome is the end product of gene transcription, which means that changes in the metabolome are amplified with regard to alterations in the transcriptome and proteome. For example, the levels of metabolites are determined by the expression of biosynthetic enzymes, making their levels a complex function of many intra- and extracellular regulatory processes along different dimensions [32]. This omic technology has been successfully applied to: (a) detection and selection of new active substances [33]; (b) detection of subtle alterations in biological pathways to provide insight into the mechanisms underlying physiological and aberrant processes [34]; (c) determination of biomarkers [31]; and (d) determination of the effects of different substances on biological systems [35]. The types of samples that can be analyzed using metabolomics are wide-ranging and include tissues, cells, and bio-fluids [36]. The analysis of the cellular metabolome is an approach for identifying specific metabolic responses to different stimuli, providing explicit biochemical information on the metabolic mechanisms. Thus, targeting a specific cell type may reduce variability and provide a more consistent background in which more subtle metabolic changes become more apparent [37]. This is particularly valuable in the case of studies conducted at the level of NDs, where the metabolomic determination in biofluids might be highly nonspecific due to the inclusion of variables such as environmental exposure, gender-specific responses, age, and genetics, among others [38].

The nature of metabolomics data presents a challenge for traditional statistical analyses and interpretation. A considerable amount of data pre-processing and preparation is required in order to analyze the information. For these reasons, integration of metabolomics data with machine learning (ML) has proven to be a powerful predictive approach, resulting in improvements in several areas such as diagnostic biomarker prediction, clinical data integration, networks, systems approaches to neural behavior, novel etiology markers for inflammation, neurodegeneration progression, and even mass spectrometry image analysis [39]. In this regard, by exploiting metabolomics data with ML, it is possible to improve the detection of new potential biomarkers or to unveil new molecular mechanisms associated with primary metabolism and lipid impairment across neurodegeneration. This study aimed to explore the metabolomic profiles of astrocytes under lipotoxic conditions in order to provide insights such as potential biomarkers for scenarios of lipotoxicity induced by PA, which have not been identified in human astrocytes and are proposed as candidates for further research and validation.

## 2. Materials and Methods

### 2.1. Cell Culture

Three different lots (#0000612736, #0000565612, #0000514417) of normal human astrocyte (NHA) primary cells (Lonza^®^, Walkersville, MD, USA; Clonetics™, NHA-Astrocytes AGM), were used as an in vitro model. This has been applied in numerous studies on neurotoxicity, neurogenesis, brain damage, and neurodegenerative disorders [40,41,42]. NHA has been characterized by immunofluorescence for GFAP (glial fibrillary acid protein) and has been observed along the passages, guaranteeing a pure culture of human astrocytes. Cells were maintained on astrocyte growth medium (ABM) (Lonza^®^) supplemented with BulletKit (Lonza^®^), containing rhEGF, 0.5 mL; insulin, 1.25 mL; ascorbic acid, 0.5 mL; GA-1000, 0.5 mL, L-glutamine, 5.0 mL; and fetal bovine serum, 15 mL. Cultures were incubated at 37 °C in a humidified atmosphere with 5% carbon dioxide. The medium was replaced three times a week.

### 2.2. Treatment of Human Astrocytes with Palmitic Acid

NHA cells were seeded in 48-well plates at 5000 cells/cm^2^ for 12 days, then washed with phosphate-buffered saline (PBS) 10X and starved in free serum DMEM without L-glutamine, phenol red, or supplements (Lonza^®^) for 6 h. PA was used to induce lipotoxic damage [12,43]. Cells were treated with 2 mM palmitic acid (PA) (Sigma-Aldrich Inc., St. Louis, MO, USA), 1.35% fatty acid-free bovine serum albumin as the carrier protein (Sigma-Aldrich A2153), 2 mM carnitine for fatty acid transport into the mitochondrial matrix, and phenol-red free, serum-free, and additive-free basal endothelial culture medium EBM (Lonza^®^) for 24 h. This PA concentration and suitable time were determined that induced 50% cytotoxicity in a sensitivity experiment comparing 6 concentrations from 100 µM to 2 mM. Based on our results, 24 h of treatment with PA at 2 mM was selected for the following experiments.

### 2.3. Endogenous Metabolite Extraction (Fingerprinting)

For the extraction of metabolites, we considered the protocols published by [35,44,45]. Briefly, culture medium of each sample was discarded, washing the cells twice with 1X PBS at 37 °C. Then 1.5 mL of methanol (HPLC grade) at −80 °C was added to arrest cell metabolism. The cells were then detached with a scraper and immediately transferred to a 2 mL Eppendorf containing 10 µL of internal standard (norvaline dissolved in pyridine). Subsequently, it was incubated in an ice bath for 10 min, and then, frozen in liquid nitrogen for other 10 min. Afterwards, samples were thawed in an ice bath for 10 min with brief vortexing and centrifuged at 4 °C, 12,000 rpm, for 5 min. The supernatant was transferred to a new tube, and extraction was carried out twice with 250 µL of cold 80% methanol. The supernatants were combined, and the cell pellet was discarded. Finally, the extracts were dried under a continuous nitrogen flow and the remaining water was lyophilized. After drying, the samples were derivatized and submitted to GC–TOF-MS at the West Coast Metabolomic Center (WCMC).

### 2.4. Exogenous Metabolite Extraction (Footprinting)

Extraction of exogenous metabolites was performed using previously described protocols [45]. Briefly, an extraction solution containing acetonitrile, isopropanol, and water (3:3:2 *v*/*v*) was prepared, degassed under nitrogen flow for 5 min, and cooled down to −20 °C. One milliliter of extraction solution was added to the dried samples, vortexed for 10 s, and transferred to the thermo-shaker for 5 min at 4 °C. The samples were then centrifuged at 14,000 rcf for 2 min. The supernatant was divided into two 500 µL fractions, one for analysis and one for backup, which were stored at −20 °C. One of the fractions was evaporated under vacuum to complete dryness. Lastly, the dried samples were resuspended in 500 µL of 50% acetonitrile solution (degassed), stirred for 10 min, and centrifuged at 14,000 rcf for 2 min; the supernatant was transferred to a new tube, where it was dried under vacuum and subsequently derivatized.

### 2.5. Derivatization

Briefly, the dried samples were spiked with 975 µL of MeOH:MTBE and QC mix. Then, 188 µL of water (LC grade) was added and shaken at 4 °C for 6 min. Subsequently the samples were centrifuged for 20 s at 14,000 g, and both phases were separated and placed in different tubes and dried using a centrivap. The upper phase was reconstituted adding 110 µL MeOH:Tol (9:1) + CUDA (50 ng/mL), vortexed (10 s), sonicated (5 min), and centrifuged (2 min at 16,100 g). Then, 45 µL was placed in an amber vial with a microinsert.

### 2.6. Metabolomics Analysis

#### 2.6.1. Instrumental Analysis

Metabolomics analysis was performed in the West Coast Metabolomics Center (WCMC). The full conditions of data acquisition are reported elsewhere [30]. Briefly, GC–TOF-MS analysis was performed using an Agilent 6890 chromatograph (Santa Clara, CA, USA) coupled with a Leco Pegasus IV mass spectrometer (St. Joseph, MI, USA). An Rtx-5Sil MS column (30 m × 0.25 mm id × 0.25 µm film, Restek Corporation, Bellefonte, PA, USA) made of 95% dimethyl/5%diphenylpolysiloxane, protected with a 10 m empty guard column was used. The mobile phase was helium with a flow rate of 1 mL·min^−1^. The injection volume was 0.5 µL in splitless mode into a multi-baffled glass liner with an injection temperature of 50 °C ramped to 250 °C by 12 °C·s^−1^. The liner was automatically changed after 10 injections. The oven temperature program was 50 °C (1 min), ramped at 20 °C·min^−1^ to 330 °C (5 min). Mass spectrometer parameters were: m/Z acquisition was from 80 to 500 Da, with unit mass resolution al 17 spectra·s^−1^, the ionization energy was set at 70 eV and 1800 V detector voltage with a 230 °C transfer line and 250 °C ion source. Every 10 samples and after the change of the liner, the fames mixture and the quality control mixture were run.

#### 2.6.2. Data Processing and Annotation

Data processing was performed using ChromaTOF v.2.32 (Leco Corporation, St. Joseph, MI, USA) with the analytical platform of the WCMC. The raw data were processed without smoothing, 3 s peak width, baseline subtraction above the noise level, and automatic mass spectral deconvolution and peak detection at signal/noise levels of 5:1 throughout the chromatogram. The BinBase algorithm and database was used to annotate the features. The parameters used for the feature selection were: the validity of the chromatogram (less than 10 peaks with intensity greater than 107 counts s^−1^), unbiased retention index marker detection (MS similarity greater than 800), and retention index calculation by 5th order polynomial regression. Spectra were cut to 5% base peak abundance.

The feature annotation was performed by matching each convoluted feature to database entries from most- to least-abundant spectra using the following filters: retention index window ±2000 units (equivalent to about ±2 s retention time), validation of unique ions and apex masses (unique ion must be included in apexing masses and present at greater than 3% of base peak abundance), and mass spectrum similarity must fit criteria dependent on peak purity and signal/noise ratios and a final isomer filter.

### 2.7. Metabolomics Data Analysis with Machine Learning

Data analysis was carried out following the flow chart shown in Figure 1. First, data tidying was performed. Every feature that presented more than 70% of zeros in all the samples was removed. Then, the remaining zeros were replaced with the minimum value (baseline intensity). Subsequently, the data were normalized using the sum of the TIC. Then, the data were transformed using a log base ten function and scaled using a pareto algorithm, as this algorithm kept the data structure partially intact [46].

Considering the high dimensionality of the untargeted metabolomics data, a principal component analysis (PCA) [47] and hierarchical clustering analysis (HCA) [48] were performed to explore data structure and behavior, seeking the tendency of the clusters between groups and metabolites, and to detect possible outliers.

After exploratory data analysis by PCA and HCA, different supervised methods were carried out to select the best model performance. These included random forest, K-nearest neighbor, and stepwise regression. For more details on methods, see Burkov (2019) [49]. Regarding model training, leave-one-out cross validation (LOOCV) was employed, which is a method more effective on smaller datasets (fewer observations in binary comparisons). Feature selection was performed through ANOVA and pairwise T-tests across all features (metabolites) with a *p*-value < 0.05. These metabolites were selected for model training and testing (not shown). Additionally, PLS-DA analysis was conducted to prioritize potential biomarkers based on their variable importance in projection (VIP) value. Model training and evaluation were tested with different packages implemented in R: “caret” (https://cran.r-project.org/web/packages/caret/caret, accessed on 10 April 2021); “Mleval” (https://cran.r-project.org/web/packages/MLeval, accessed on 10 April 2021); “class” (https://cran.r-project.org/web/packages/class, accessed on 10 April 2021); and “pROC” (https://cran.r-project.org/web/packages/pROC/pROC, accessed on 10 April 2021).

Upon ML selection, further analyses were applied using the MetaboAnalyst tool [50] (https://www.metaboanalyst.ca/, accessed on 10 April 2021) for biological interpretation of model results. Enrichment and pathway analysis were performed employing the final metabolites selected. Results provided enrichment and impacts of several pathways (S1), which were further analyzed for biological significance and potential biomarker identification.

## 3. Results and Discussion

In this study, we employed three state-of-the-art algorithms, namely, random forest, K-nearest neighbor, and stepwise regression, to obtain high accuracy models to predict metabolites associated with lipotoxicity induced by PA as potential biomarkers. Our study showed that the best model was based on random forest. This ML model yielded a 93% area under the ROC curve, with sensibility and specificity values of 80% and 93%, respectively (Table 1). Figure 2 shows the metrics of the model, evidencing robust model performance by the amount of data involved. Model quality is given by the parameters R2 (fraction of variance explained by a component) and Q2 (fraction of the total variation predicted by a component). Q2Y was above 0.75 which indicates a good predictive power of the model, and the R2Y was near 1, suggesting that the total variation of the data is mainly explained by the components in the model. 

Following the extraction of the most important variables (metabolites) from the random forest classifier, ten of these metabolites were identified and are reported in Figure 3. The highest relevance metabolites that presented upregulation were alanine, adenosine, citramalic acid, citric acid, ethanol phosphate, glutamine, glyceric acid, palmitic acid, sucrose, and threonic acid. It is worth mentioning that the models created based on the logistic regression and stepwise regression algorithms also showed the 10 most important variables, shown in Figure 3, with some differences in the weights of each one, which demonstrates the value and potential of the machine learning methods in this context. Through ML model and enrichment analyses, we identified the seleno-amino acid metabolism, urea cycle and nitrogen metabolism, glutamate and glutamine metabolism, and the alanine metabolism pathways with the highest rates of enrichments, supported by pathway impact and significative *p*-values. Importantly, there were still a number of unknown features with high variable significance, most likely related to the previously mentioned pathways, making then potentially useful contributors to the understanding of the underlying mechanisms of lipotoxicity.

Our results were shown to be highly correlated with multiple brain disorders and related diseases such as hyperlipidemia in rats [51], hyperammonemic syndrome [52], macrophage lipotoxicity [53], ischemia [54], brain tumors [55], and PD [56], among others. Likewise, most of these pathways may be linked to common neurodegeneration hallmarks such as oxidative stress [57], inflammation and trauma [54], and other types of stressful insults, as will be discussed below.

Seleno compounds, which include several of our final metabolites proposed as potential biomarkers, such as alanine and adenosine, have been reported to be factors that elicit cellular responses, generating cytotoxic effects on cells [58], but the mechanisms are still unknown. Importantly, glutathione peroxidase has been reported to be a major protective enzyme against oxidative damage in the brain [59], since it decreases hydrogen peroxide and FA hydroperoxides. As previously mentioned, alanine is an important metabolite in seleno compound metabolism, and alanine-induced cytotoxic oxidative stress in brain tumor cells has been reported [55]. It has been reported that alanine is a substrate for producing hydrogen peroxide. In the results obtained in this study, we found an upregulation of alanine, which may cause increased lipid peroxidation. However, the enzyme d-amino acid oxidase (DAO) has been reported to provide protection against the cytotoxic effects of alanine. It has also been suggested that alanine toxicity is impaired under conditions of glutathione peroxidase depletion [55]. Interestingly, researchers have reported that these findings, when combined with simulation of pentose phosphate pathway (PPP) activity, show even greater expression of cytotoxicity in tumor cells [55]. Though PPP has not been identified or related to lipotoxicity in the present study, previous transcriptomic research by our laboratory using the same human astrocytes with PA-induced lipotoxicity suggest that PPP is one of the main pathways affected [60]. This evidence highlights alanine as a mediator of cytotoxicity and oxidative stress, supporting the data shown by our model, making it a potential biomarker having an important impact on lipotoxicity in human astrocytes, as shown in Figure 4.

As mentioned above, the enzyme glutathione peroxidase-1 (GPX1) is of great importance in oxidative stress conditions, because it represents the first identified mammalian seleno-protein [59]. In addition, multiple regulators of GPX1 activity have been described. One of these regulators is adenosine, a metabolite that has been highlighted by the model and that shows significant dysregulation; it has been proposed to be protective against ROS by reducing oxidative damage in mice cells [59]. In another study, treatment of primary human pulmonary microvascular endothelial cells (HPMECs) with adenosine in the presence of an adenosine deaminase inhibitor resulted in elevated GPX1 mRNA levels and enzyme activity [59]. In contrast, a study performed by Manzoni et al. (2020) introducing lipotoxicity through hyperlipidemia in rats [51], reporting that hyperlipidemia increased uric acid levels, which may be directly correlated with the adenosine deaminase (ADA) activity, since uric acid is a product of adenosine metabolism. Moreover, adenosine has several receptors through which it may impart immune and regulatory actions. Some of these receptors are activated by elevated adenosine concentration conditions often found in ischemia, trauma, and inflammation scenarios [54]. In this study, an upregulation of adenosine was shown by the model, supporting the idea that increased levels of adenosine can be related to lipotoxic conditions by increasing immune regulation via these receptors. Altogether, adenosine plays a pivotal role in the regulation of stressful insults in cells and may be proposed as a major factor in human astrocyte lipotoxicity with the mentioned mechanisms shown in Figure 4.

Another important finding is the role of glutamine and glutamate metabolism as a key factor in lipotoxicity. Glutamate metabolism has largely been associated with many neurodegeneration conditions and brain disorders [61,62,63,64]. Interestingly, urea metabolism is closely related to glutamine and both play crucial roles in hyperammonemic syndromes. Since the urea cycle is mainly responsible for ammonia removal, and glutamine synthesis is attributed to astrocytes, it is clear that in conditions of hyperammonemia, the excess of glutamine may affect astrocyte morphology and function [52]. Indeed, the increase of glutamine concentration represents a hallmark for hyperammonemia, which may be followed by several neurotoxic responses such as astrocyte swelling and cerebral edema [52] (Figure 4). The relation between these pathways has been largely proven, and urea cycle disorders often result in alterations of the neurotransmitter systems [65], which is expected, as glutamine synthase is the main enzyme in the brain for ammonia detoxification [65]. More specifically, glutamine can elicit responses in a number of ways, acting as an idiogenic osmole, dysregulating K homeostasis, altering oxidative metabolism, and therefore releasing free radical species [52] (Figure 4). Another impact of abnormal glutamine levels in the brain is shown through macrophage activation. Glutamine has been reported to modulate macrophage lipotoxicity [53], which is well-known to be a trade-off condition in obesity and diabetes [3,4]. On the other hand, studies have proposed that in an excess of lipids, glutamine metabolism could overwhelm mitochondria and promote the accumulation of toxic metabolites [53] such as alanine. More recently, findings indicate that glutamine deficiency may reduce lipid-induced lysosome dysfunction, inflammation, and cell death [53]. In this sense, glutamine deficiency prevents the suppressive effect of PA on respiration, supporting the already-accepted idea of toxicity induced by increased levels of glutamine.

Citric acid cycle flux has been reported to be upregulated by PA, which also alters β-oxidation in mitochondria in neonatal rat cardiomyocytes [66]. The results obtained by this 2016 study indicate that this metabolic failure occurs prior to cell death, suggesting that it might be a side effect of lipotoxicity, most likely due to diacylglycerol (DAG)-mediated protein kinase C (PKC) activation rather than oxidative stress [66], interestingly matching the dysregulation shown by the results of our study. However, further research and study is necessary in order to determine the specific cause. Citric acid is another final metabolite reported by the model, and hence its upregulation in human astrocytes is proposed as another potential biomarker of the lipotoxic condition. Another recognized metabolite found in the model is citramalic acid, and even though there was no evidence reported that it could be related to lipotoxicity or neurodegeneration, it is a metabolite highly associated with both the citric acid cycle through cis-aconitate and glutamine metabolism, indicating a potential relationship with the above-mentioned pathways and metabolites, which could be a promising metabolite for further research and targeting in a neurodegenerative context.

In astrocytes, the signaling system has an important influence on neurons. For example, alanine is released to serve as a nitrogen donor for the synthesis of amino acids and nitrogenous metabolites. The glutamate and d-serine synthesized and released from astrocytes enhance N-methyl-d-aspartate (NMDA) receptor-mediated current, and the dysregulation produces failure in excitatory feedback to neurons [67]. Moreover, astrocytes release the major source of ATP in the nervous system, which is converted to extracellular adenosine and acts on the presynaptic adenosine A_1_ receptors, which result in inhibition of synaptic transmission. Thus, adenosine acts as an endogenous neuroprotectant and anticonvulsant, but the dysregulation of this biomarker could be a signal of neurodegenerative and neurological disease [68,69].

Finally, one study suggests that increased threonic acid is a biomarker of PD [56]. The insights revealed that threonic acid had elevated concentrations in PD patients and correlated these results primarily with oxidative stress. Like citramalic acid, this metabolite, along with the undiscovered metabolites, represents a potential and interesting candidate to investigate and might aid in the understanding of the complex underlying mechanisms of these brain disorders and pathologies.

## 4. Conclusions

Metabolomic analysis of human astrocyte cultures exposed to PA identified potential biomarkers for lipotoxicity in a neurodegenerative context. Dysregulation in seleno-amino acid metabolism, urea cycle, and glutamate metabolism pathways are proposed to be major actors in astrocyte lipotoxic scenarios. Additionally, increased metabolites such as alanine, adenosine, and glutamine are suggested as potential biomarkers for lipotoxicity conditions in human astrocytes, supporting previous studies on mice brain, tumor cells, macrophages, and other cell types. Notably, other metabolites with less research on the topic are also shown as putative biomarkers for neurodegeneration. However, all metabolites and pathways mentioned in this study require proper validation and further research in order to make them actionable. Conversion of peak intensity values into concentrations would facilitate the construction of much more accurate models and provide specific clues on potential biomarkers for future complementary studies. Further research and experimental design for validation of these potential biomarkers is encouraged in order to elucidate how these pathways and metabolites impact on pathological scenarios through specific mechanisms, leading to actionable results such as serums with concentration ranges for disease diagnoses. 

## Figures and Tables

**Figure 1 biomolecules-12-00986-f001:**
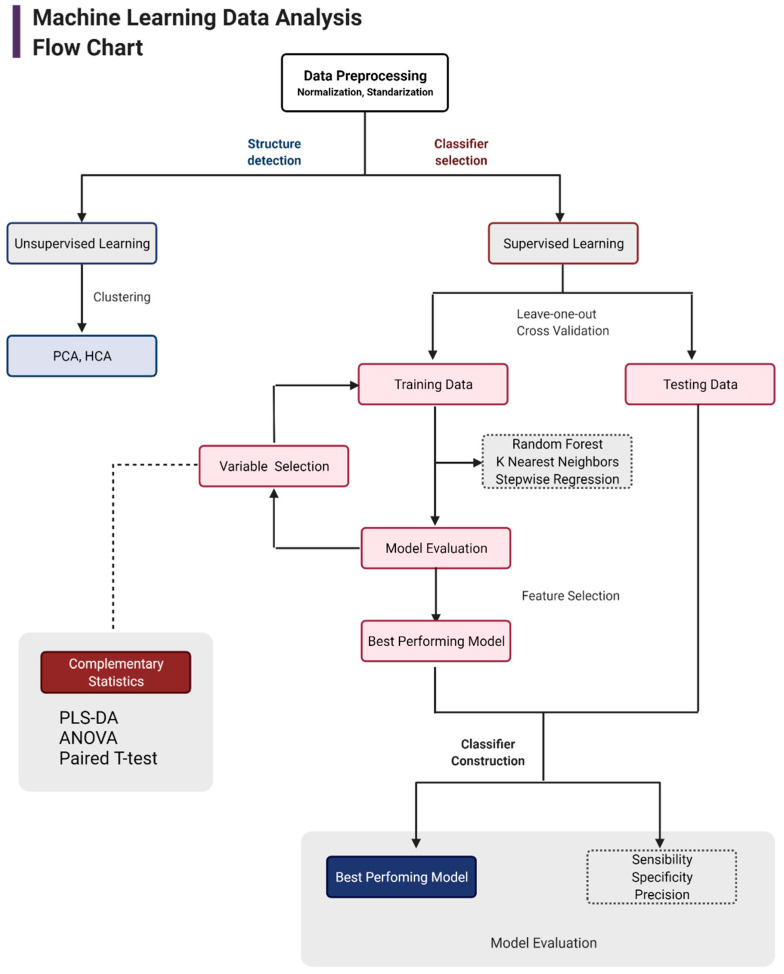
Flowchart for machine learning data analysis. This figure provides an overview of the approach used for model construction and data analysis. The left (structure detection) side indicates the unsupervised analysis, and the right (classifier selection) side indicates the supervised analysis and model construction process. Finally, the grey boxes at the bottom depict feature selection and final model evaluation parameters.

**Figure 2 biomolecules-12-00986-f002:**
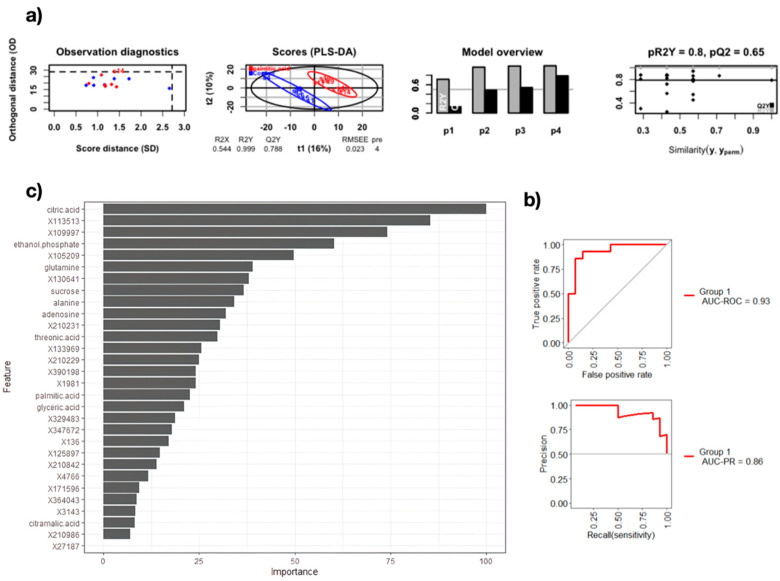
Feature selection and model performance. (**a**) PLS–DA analysis for comparing palmitic acid and control treatments from which several metabolites were extracted as features for further modeling. (**b**) Random forest ROC and precision recall curves. (**c**) Most important features (metabolites) of the random forest model. The figure was created using R software version 4.0.5. R2 means fraction of variance explained by a component and Q2 means fraction of the total variation predicted by a component.

**Figure 3 biomolecules-12-00986-f003:**
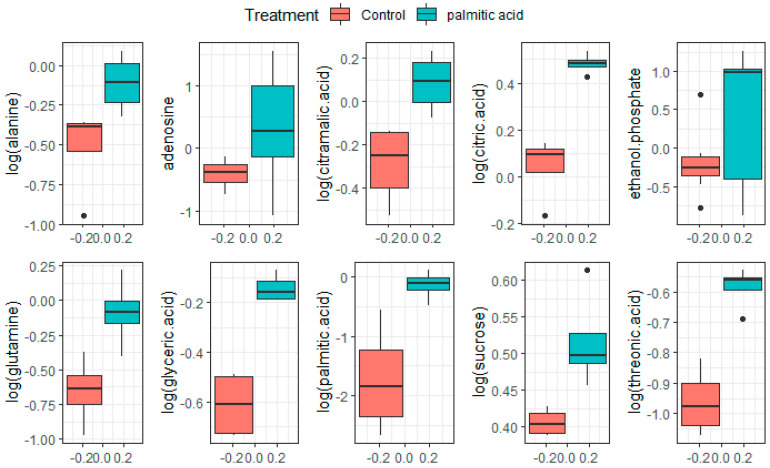
Boxplots of normalized peak intensity values of final known metabolites obtained from the random forest model for control and PA treatments. The data are shown in alphabetic order. The figure was created using R software. Bold dotes denote outliers.

**Figure 4 biomolecules-12-00986-f004:**
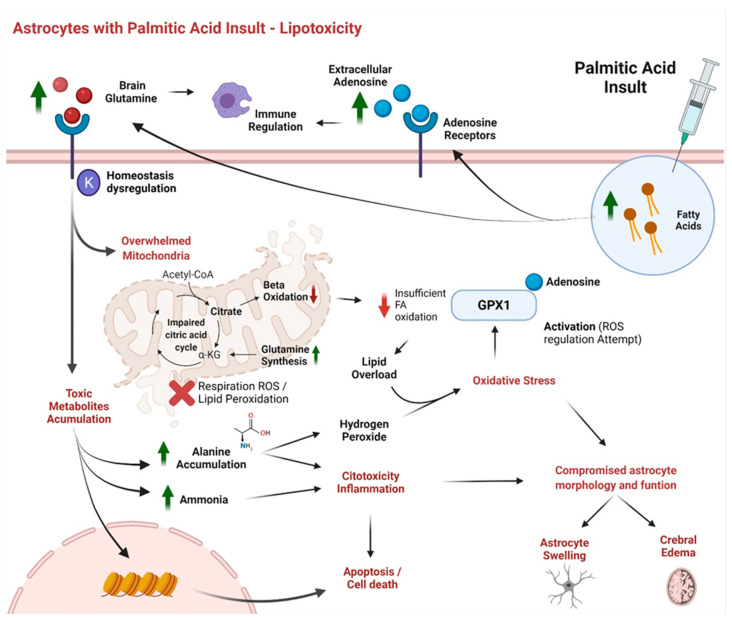
Metabolites and pathways with biomarker potential for lipotoxicity induced by palmitic acid in human astrocytes. Accumulation of alanine, adenosine, citric acid, and glutamine are proposed as biomarkers for lipotoxicity in human astrocytes. Principal pathways involved comprise seleno-amino acid compounds, glutamine and glutamate metabolism, the citric acid cycle, the urea cycle, and alanine and adenosine metabolism. The figure highlights their potential activities in astrocyte metabolism and the consequences of dysregulation of the mentioned metabolites, resulting in scenarios related to lipotoxicity such as cytotoxicity, inflammation, apoptosis or cell death, astrocyte swelling, and cerebral edema.

**Table 1 biomolecules-12-00986-t001:** Benchmark of model performance among ML methods for classifier construction. Algorithms employed were random forest, K-nearest neighbor, and stepwise regression. Metrics evaluated consisted of the area under the ROC curve, sensibility, and specificity. The best performing method was random forest, clearly over-performing the remaining algorithms.

Metric	Random Forest	K-Nearest Neighbor	Stepwise Regression
Sensibility (%)	80.57	92.85	92.85
Specificity (%)	85.71	85.71	71.42
Area under the ROC curve (%)	93.06	93.62	79.84
Area under the precision recall curve	86	44	25

## Data Availability

Not applicable.

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
