# Peer review of "Metabolomic Analysis of Human Astrocytes in Lipotoxic Condition: Potential Biomarker Identification by Machine Learning Modeling"

_biomolecules, 2022, doi:10.3390/biom12070986_

Round 1

Reviewer 1 Report

In the study by Baez et al. titled “Metabolomic Analysis of Human Astrocytes in Lipotoxic Condition: Potential Biomarker Identification by Machine Learning Modeling”, the authors employed algorithms Random Forest, K-Nearest Neighbors and Stepwise regression to obtain high accuracy models to predict metabolites associated with lipototoxity induced by Palmitic acid on astrocytes. The authors conclude that Random Forest method is better in comparison to the other two methods. There are several articles that directly address the effect of PA on astrocytes for eg., Ortiz-Rodriguez et al., 2019; Dobri et al., 2022; Gupta et al., 2012 and show difference in cytokine levels, induction of autophagy and so on. Moreover, the team itself has published machine learning algorithms and prediction of metabolites and transmitters in astrocytes. One of the main drawbacks of ML is reproducibility, which the authors also admit. Therefore, the significance of the study is minimal. I have few other comments to the authors.

Abstract:

The abstract is too lengthy and must be significantly shortened.

Introduction:

The authors consistently use Seleno-Amino acids, which is incorrect. Selanoproteins are proteins rich in Selenium containing amino acid such as selenocysteine amino acid or selenomethionine. While glutathione peroxidases are selanoproteins, the authors use the term seleno-glutathione peroxidase, which is incorrect.

Discussion:

The discussion is exhaustive. But the main issue is that the authors fail to discuss the relevance of their results with respect to astrocytes, i.e,what is the significance of the specific metabolic biomarkers on Astrocytes upon PA treatment. This is critical because the study is performed to understand the influence of Palmitic acid on astrocytes. The current discussion is descriptive and includes other cell types such as endothelial cells or myocytes.

Line 152-154: Missing reference

Figure 3: Figure is cropped and is missing portions.

Line 412: What is the relevance of Tibolone to this study. The line should be removed.

Minor comments:

Line 145: Please write full name Palmitic acid

Line 147: 10XPBS?

Line 149: Include abbreviation PA in

Author Response

Manuscript ID: Biomolecules-1733356

Manuscript title: Metabolomic Analysis of Human Astrocytes in Lipotoxic Condition: Potential Biomarker Identification by Machine Learning Modeling

Dear Dr. Celia Sun
Assistant Editor
e-mail:  celia.sun@mdpi.com

We are pleased to resubmit the research paper entitled “Metabolomic Analysis of Human Astrocytes in Lipotoxic Condition: Potential Biomarker Identification by Machine Learning Modeling” after the changes requested by reviewers for Biomolecules. We are very grateful to the reviewers for their comments and we have incorporated all the constructive considerations in this new and improved version of the manuscript. We have replied to each of the points raised by the reviewers. We hope reviewers find all their concerns addressed. Please find below all of our responses and changes made to the manuscript.

Reviewer 1

Comment 1:

  • Abstract: The abstract is too lengthy and must be significantly shortened.

Response:

 The abstract had 361 words it has now been reduced to 290, Reducing it by 20%

  • Recently, the association between neurodegenerative diseases (NDs) and obesity has been well studied. Obesity is a syndrome of multifactorial etiology characterized by an excessive accumulation and release of fatty acids (FA) in adipose and non-adipose tissue. Excess of FA generates a metabolic condition known as lipotoxicity, which triggers pathological cellular and molecular responses causing dysregulation of homeostasis and decrease in cell viability. This condition is a hallmark of NDs and astrocytes are particularly sensitive to it, given their crucial role in energy production and oxidative stress management in the brain. However, analyzing cellular mechanisms associated with these conditions represents a challenge. In this regard, metabolomics is an approach that allows biochemical analysis from a comprehensive perspective of cell physiology. This technique allows determining cellular metabolic profiles in different biological contexts such as those of NDs and specific metabolic insults such as lipotoxicity. Since data provided by metabolomics can be complex and difficult to interpret, alternative data analysis techniques such as Machine Learning (ML) have grown exponentially in areas related to omics data. Here, we developed a ML model yielding a 93% area under the Receiving Operating Characteristic (ROC) curve, with a sensibility and specificity values of 80% and 93%, respectively. This study aimed to analyze the metabolomic profiles of human astrocytes under lipotoxic conditions to provide powerful insights such as potential biomarkers for scenarios of lipotoxicity induced by Palmitic Acid (PA). In this work, we propose that dysregulation in seleno-amino acid metabolism, urea cycle and glutamate metabolism pathways are major triggers in astrocytes lipotoxic scenarios, while increased metabolites such as Alanine, Adenosine and Glutamate are suggested as potential biomarkers, which, to our knowledge, have not been identified in human astrocytes and are proposed as candidates for further research and validation.”

Comment 2:

  • Introduction: The authors consistently use Seleno-Amino acids, which is incorrect. Selanoproteins are proteins rich in Selenium containing amino acid such as selenocysteine amino acid or selenomethionine. While glutathione peroxidases are selanoproteins, the authors use the term seleno-glutathione peroxidase, which is incorrect.

Response:

We have removed the term seleno-glutathione peroxidase and specified that it is a seleno protein instead.

For the use of the term: “Seleno amino acids”, we found that the use of seleno amino acids is widely used in the literature and also in important Metabolic repositories and data bases such as:

  • PubChem, NIH: https://pubchem.ncbi.nlm.nih.gov/pathway/Reactome:R-HSA-2408522
  • The reactome: https://reactome.org/PathwayBrowser/#/R-HSA-2408522&DTAB=MT

Also it has been reviewed by journals such as Frontiers and Sciencedirect.

  • https://www.frontiersin.org/articles/10.3389/fpls.2022.804368/full
  • https://www.sciencedirect.com/topics/agricultural-and-biological-sciences/seleno-amino-acids

Comment 3:

  • Discussion: The discussion is exhaustive. But the main issue is that the authors fail to discuss the relevance of their results with respect to astrocytes, i.e,what is the significance of the specific metabolic biomarkers on Astrocytes upon PA treatment. This is critical because the study is performed to understand the influence of Palmitic acid on astrocytes. The current discussion is descriptive and includes other cell types such as endothelial cells or myocytes.

Response:

The discussion of the metabolites in astrocytes was included from the line 389 to 398

Comment 4: Line 152-154: Missing reference

Response: References are included

Comment 5: Figure 3: Figure is cropped and is missing portions.

Response:

The figure has been fixed and replaced in the manuscript, as shown below.

Comment 6: Line 412: What is the relevance of Tibolone to this study. The line should be removed.

Response:

The line has been removed

Comment 7 Line 145: Please write full name Palmitic acid

Response: The full name has been written as shown below:

  • “2.2. Treatment of human astrocytes with Palmitic Acid”

Comment 8: Line 147: 10XPBS?

Response: the expression was corrected as shown below:

washed with Phosphate-Buffered Saline (PBS) 10X

Comment 9: Line 149: Include abbreviation PA in

Response:

The abbreviation has been included as shown below:

  • “Treated with 2 mM palmitic acid (PA) (Sigma-Aldrich, P5585), 1.35% fatty acid-free bovine”

We thank you in advance for considering our manuscript for publication in Biomolecules.

Best wishes,

The authors

Reviewer 2 Report

This is a very interesting and relevant study in which the authors exposed normal human astrocytes in culture to a free fatty acid (FA) palmitic acid (PA) and then extracted the cells with a water-acetonitrile solvent and analyzed the extraction by GC with mass spectrometer (MS) detection. They then applied "machine learning" (ML) to the MS detector output and examined three standard paradigms for ranking data: 1. random forest; 2. K-nearest neighbors; and 3. stepwise regression. The goodnesses of fit of each ML model to the data were determined by 1. sensibility (aka sensitivity); 2. specificity; 3. area under the receiver operating characteristic (ROC) curve; and 4. area under the precision recall curve. Based on these models, the authors chose the "random forest" model to identify 10 metabolites. They provide data from their GC-MS analyses that these 10 metabolites are all elevated in astrocytes exposed to palmitic FA and propose several metabolites as potential biomarkers of "lipotoxicity".

This paper is relevant b/c lipotoxicity to astrocytes may be an underlying mechanism in neurodegeneration, and their method may contribute to searches for biomarkers of astrocyte lipotoxicity. However, there is one major limitation which is easily addressed. The authors need to show the readers which metabolites emerge from the other two paradigms used (K nearest neighbor and stepwise regression- like in Fig 2c), and how these metabolites compare to the control and PA treated cells. Doing this would strengthen the authors' argument that ML can contribute to the lipotoxicity biomarker search.

Otherwise the paper is well written and can make a meaningful contribution to the literature on brain lipotoxicity.

Author Response

Manuscript ID: Biomolecules-1733356

Manuscript title: Metabolomic Analysis of Human Astrocytes in Lipotoxic Condition: Potential Biomarker Identification by Machine Learning Modeling

Dear Dr. Celia Sun
Assistant Editor
e-mail:  celia.sun@mdpi.com

Biomolecules journal

We are pleased to resubmit the research paper entitled “Metabolomic Analysis of Human Astrocytes in Lipotoxic Condition: Potential Biomarker Identification by Machine Learning Modeling” after the changes requested by reviewers for Biomolecules. We are very grateful to the reviewers for their comments and we have incorporated all the constructive considerations in this new and improved version of the manuscript. We have replied to each of the points raised by the reviewers. We hope reviewers find all their concerns addressed. Please find below all of our responses and changes made to the manuscript.

Comment

  1. The authors need to show the readers which metabolites emerge from the other two paradigms used (K nearest neighbor and stepwise regression- like in Fig 2c), and how these metabolites compare to the control and PA treated cells. Doing this would strengthen the authors' argument that ML can contribute to the lipotoxicity biomarker search.

Response:

  • We have included in the text that all 3 machine learning algorithms have shown the 10 metabolites shown in figure 3 as VIP variables obtained from the model, having the RF model yielding the best results, as shown below in line 279 to line 282 :

  • “It is worth mentioning that the models created based on the logistic regression and stepwise regression algorithms also show the 10 most important variables shown in Figure 3, with some differences in the weights of each one, which demonstrates the value and potential of the machine learning methods in this context”

We thank you in advance for considering our manuscript for publication in Biomolecules.

Best wishes,

The authors

Round 2

Reviewer 2 Report

The authors addressed my original concern about the output of the other two algorithms used in their Machine Learning approach to the GC-MS data from their experiments. Their addressing my prior concerns is noted in lines 279-282 of the appendix. They have also substantially revised the paper with new text (indicated by the yellow highlighting ). Overall the paper is much improved. Although there are a few minor grammatical and spelling errors, these are easily addressed by careful editing.

I feel that it now can be published